# Efficacy and safety of a new ranibizumab biosimilar CKD-701 using a pro re nata treatment regimen in neovascular age-related macular degeneration: A phase 3 randomized clinical trial

Chang Ki Yoon[1]☯, Jaeryung Oh[2]☯, Kunho Bae[1], Un Chul Park[1], Kyung-Sang Yu[3], Hyeong Gon Yu[1]*

1 Department of Ophthalmology, Seoul National University College of Medicine, Seoul, South Korea,
2 Department of Ophthalmology, Korea University College of Medicine, Seoul, South Korea, 3 Department of Clinical Pharmacology and Therapeutics, Seoul National University College of Medicine and Hospital, Seoul, South Korea

☯ These authors contributed equally to this work.
* hgonyu@snu.ac.kr

## Abstract

### Purpose

This study aimed to establish the efficacy, safety, and immunogenicity equivalence of the proposed biosimilar CKD-701 with the reference ranibizumab in patients with treatment-naïve neovascular age-related macular degeneration (nAMD).

### Patients and methods

A total of 312 participants with active subfoveal choroidal neovascularization were randomly assigned to either the CKD-701 (n = 156) or reference ranibizumab (n = 156) arms. The initial 3-month loading intraocular injections were followed by pro re nata (PRN) dosing for 9 months. The primary outcome was the proportion of patients with less than 15-letters of corrected visual acuity (BCVA) loss at 3 months visit (one month after last loading injection) compared to the baseline time point. The presence of retinal fluid, and changes in BCVA and central retinal thickness (CRT) were assessed as secondary efficacy outcomes. Immunogenicity and safety were evaluated in both treatment arms.

### Results

In the CKD-701 arm, 143 (97.95%) patients lost <15 letters in the BCVA at 3 months compared to 143 (98.62%) in the reference arm ($P = 0.67$). The BCVA improved with a mean improvement of +7.0 (CKD-701) and +6.2 (ranibizumab) letters at 3 months ($P = 0.43$). The least-squares mean (SE) changes in CRT at 3 months from the baseline were −119.3 (12.0) μm and −124.5 (11.9) μm in the CKD-701 and ranibizumab groups, respectively ($P = 0.74$). The proportion of participants with subretinal or intraretinal fluid at 3, 6, and 12 months was

**Data Availability Statement:** All relevant data are within the paper and its Supporting information files.

**Funding:** The study was funded by the sponsor, CKD. Ltd (Chong Kun Dang). Study design, collection and analysis of data were performed by the respective groups of the sponsor. Korean Association of Retinal Degeneration also funded this study and participated as image reading center (Seoul Reading Center). The funders had no role in study design, data collection and analysis, decision to publish, or preparation of the manuscript.

**Competing interests:** The authors have declared that no competing interests exist.

similar between the study arms. The number (SE) of injections were 8.36 (3.13) in the CKD-701 and 8.26 (2.92) in ranibizumab ($P$ = 0.62). The occurrence of adverse events and anti-drug antibody in the study arms were also not statistically different.

## Conclusion

CKD-701 is a biosimilar to the reference ranibizumab in terms of efficacy, safety, and immunogenicity for the treatment of patients with nAMD. Moreover, improvement and maintenance of visual outcome were achieved through PRN regimen.

## Introduction

Age-related macular degeneration (AMD) is the most common cause of permanent blindness among the elderly in developed countries [1], as the prevalence of macular degeneration increases with age. The number of patients with neovascular AMD (nAMD) has increased owing to the rapidly aging population [2]. Intraretinal or choroidal neovascularization (CNV) is a hallmark of nAMD, and the vascular endothelial growth factor (VEGF) is the primary target for nAMD treatment [3]. Therapeutic agents for wet AMD, such as ranibizumab, aflibercept, and brolucizumab block the VEGF, which plays an important role in angiogenesis [4]. Ranibizumab is a humanized murine anti-VEGF-A monoclonal antibody fragment that binds to and neutralizes the active isoforms of VEGF-A. It was first approved for wet AMD in the United States in 2005 and the clinical indications have expanded from AMD to diabetic retinopathy, retinal vascular occlusion disease, and myopic CNV [5–7]. As a result, the clinical demand for anti-VEGF agents is increasing [8].

Since the patents for these drugs have recently expired, biosimilars are therefore being developed [9–11]. The relatively high cost of the original biological agents likely limits some patient access to these treatments; by contrast, biosimilar agents can help reduce costs and improve patient accessibility. For example, Razumab™ (Intas Pharmaceuticals Ltd., Ahmedabad, GJ, India), a biosimilar to innovator ranibizumab, has gained marketing approval (in 2015) in India for nAMD, diabetic macular edema, and retinal vein occlusion [8,9]. In addition, several phase 3 clinical trials for ranibizumab or other aflibercept biosimilars have been completed or are in progress [9,12,13].

Biosimilars are biological medicinal products that contain a version of the active substance of an already authorized original biological medicinal product (reference medicinal product) [14]. Although Biosimilars are similar in terms of quality characteristics, biological activity, safety, and efficacy based on a comprehensive comparability exercise, unlike chemical drugs, their pharmacological properties cannot be assumed to be the same because the entire molecular structure of biosimilars may not be identical to the reference [14]. CKD-701 is a novel candidate biosimilar to ranibizumab. The present study is a phase 3 clinical trial conducted to investigate the similarity in efficacy and safety between CKD-701 and the reference ranibizumab in patients with nAMD.

## Materials and methods

This was a randomized, double-blind, parallel group, comparative multicenter, phase 3 clinical trial to evaluate the efficacy, safety, pharmacokinetic (PK) properties, and immunogenicity of the test biosimilar CKD-701 with the reference ranibizumab in patients with nAMD. The

study was conducted in compliance with the protocol approved by the Ministry of Food and Drug Safety of Korea and in accordance with the Declaration of Helsinki and Korean Good Clinical Practices. The clinical study protocol and protocol amendments were reviewed and approved by an independent ethics committee or the institutional review board at each clinical site. This clinical trial was registered in Korean Ministry of health and welfare as well as clini-caltrial.gov at the beginning, but the registration to the clinicaltrial.gov was delayed due to technical errors. (NCT04857177). The authors confirm that all ongoing and related trials for this drug/intervention are registered. A written informed consent form was obtained by each patient before entering the study.

This study was conducted in 25 hospitals across Korea. Participants were >50 years old with newly diagnosed anti-VEGF treatment-naïve active subfoveal CNV secondary to AMD in the study eye. The total neovascular lesion area (including neovascularization, blood, and scar) was <12 disc areas, of which at least 50% were active CNV lesions. Eligibility was confirmed through a masked assessment of fluorescein/indocyanine green angiography and optical coherence tomography images using a centralized and independent image reading center (Seoul Reading Center). Additionally, 78–34 letter score (20/32 to 20/200, Snellen equivalent) of best-corrected visual acuity (BCVA) was required in the study eye (Early Treatment Dia-betic Retinopathy Study [ETDRS] letters). The key exclusion criteria included prior intravitreal anti-VEGF treatment in either eye; history of verteporfin photodynamic therapy, laser photo-coagulation, pars plana vitrectomy, or other surgical intervention for AMD in the study eye; history of intravitreal corticosteroid therapy or intravitreal device implantation within 6 months before screening in the study eye; ocular disease other than AMD such as infection, uveitis, diabetic retinopathy in the study eye or fellow eye, which can influence the treatment intervention efficacy. Detailed inclusion and exclusion criteria are described in study protocol and patients undergoing systemic infection (S1 File).

After confirmation of eligibility, patients were randomly assigned in a 1:1 ratio to the CKD-701 arm or the reference ranibizumab arm. Stratified randomization is performed based on the presence of polypoidal choroidal vasculopathy to subjects through the Interactive Web Response System. Participants, investigators, and site personnel remained blinded throughout the study, except for the IP injection investigator and the staff assigned to maintain the masked records and codes for medications, and those who were responsible for the release of medica-tions and maintenance of the logs. Participants received 0.5 mg (0.05 mL of 10 mg/mL solu-tion) of either CKD-701 or the reference ranibizumab (through intravitreal injection for a 12-month treatment period. The treatment schedule consisted of an initial monthly dosing phase for 3 months, followed by the PRN treatment phase until 12 months. The PRN injection criteria were as follows: 1) decreased visual acuity (ETDRS) by more than five letters compared to the baseline (visit 2, baseline) and the previous highest visual acuity during the study period, 2) presence of sub or intraretinal fluid (including new or remaining cases), 3) presence of sub-or intraretinal hemorrhage (including new or remaining cases), 4) identification of new CNV, and 5) central retinal thickness (CRT) increase by $\geq 50$ μm compared to the previous lowest value. After 3 months of administration (loading phase), the clinical trial drug was re-adminis-tered when any one of the PRN administration criteria were met.

The primary objective was to compare the efficacy of intravitreal injections of the ranibizu-mab biosimilar CKD-701 against the reference product in terms of preventing vision loss, as determined by the proportion of patients who lost fewer than 15 letters of visual acuity at 3 months compared to the baseline value. The baseline time point means the time of randomiza-tion. Efficacy assessment was performed based on BCVA as assessed using the ETDRS chart. Secondary endpoints included the proportion of participants who lost <15 letters at 6 and 12 months; who lost >15 letters at 3, 6, and 12 months; who had changes in BCVA from the

baseline at 3, 6, and 12 months; who had changes in CRT from baseline at 1, 3, 6, and 12 months; and the presence of intraretinal or subretinal fluid at 3, 6, and 12 months. CRT value was recalculated after manual modification of segmentation in OCT B-scans by independent retina experts of image reading center if automatic segmentation of device was improper.

Serum concentrations of CKD-701 and Ranibizumab at pre-dose and at 1, 7, 30, 60, and 90 days after the first injection were used for PK analysis, and PK parameters were calculated through a non-compartmental method using Phoenix WinNonlin® version 8.0 (Certara, St. Louis, MO, USA). The primary PK parameters were maximum plasma concentration ($C_{max}$) and the area under the concentration–time curve from time zero to the last quantifiable time point (AUClast). The AUClast was calculated using the linear up-log down trapezoidal method. The immunogenicity evaluation included the development of anti-drug antibodies (ADAs) and neutralizing antibodies. The safety assessment included ophthalmic and systemic adverse events, vital signs, and laboratory examinations.

In the pivotal study of ranibizumab, the study group of ranibizumab 0.5 mg was compared to the placebo control group, the difference between ranibizumab 0.5 mg and the placebo group was estimated 30% (95% CI: 23%, 37%) [15]. Based on these results, the equivalence margin of this clinical trial was set to 11.5% by applying the 50% rule to the lower confidence interval (23%). Assuming a significance level of 5%, a power of 80% and 1drop-out rate of 15% after 3 consecutive injections, the number of cases required for analysis is 155 per each group.

Statistical analyses were performed using the SAS software (version 9.4, SAS Institute). In terms of "bias toward the null," which may accidentally prove equivalence from the underestimation of the treatment effect due to low compliance in clinical trials, the main analysis group was assigned as the per-protocol (PP) set and the auxiliary analysis group was assigned as the full analysis set (FAS). The validity data were then analyzed. The proportion of patients with visual loss <15 letters in the BCVA at 3 months compared to the baseline, which is the primary efficacy endpoint, was presented as the number of participants and the proportion (%) between the treatment groups. The equivalence was demonstrated when the 95% confidence interval of the difference in the treatment groups was within the equivalence limits of -11.5% and 11.5%. For the secondary efficacy evaluation variable, descriptive statistics between the administration groups were presented according to each time point. A *P* value less than .05 was considered significant.

## Results

In this study, a total of 312 participants (156 patients in the biosimilar CKD-701 test arm and 156 patients in the reference ranibizumab arm) were randomized in 25 centers across Korea (Fig 1). Participants recruitment and follow-up were conducted from September 2019 to March 2021. A total of 310 randomized participants received at least one dose of the study medication (156 patients in the test arm and 154 patients in the reference arm) and were included in the safety analysis set. Of the 310 dosed participants, 19 discontinued the loading phase, with major protocol deviation. Therefore, the PP population for this study only included 291 participants (146 patients in the CKD-701 arm and 145 patients in the reference arm). Of the 146 participants included in PP population from biosimilar arm, 57 (39.04%) were females and 89 (60.96%) were males. The mean (SD) age of these patients was 72.2 (8.09) years. In the reference product arm, of the 145 participants in the PP population, 64 (44.14%) were females and 81 (55.86%) were males, with a mean (SD) age of 71.9 (8.79) years. The demographic characteristics of the patients enrolled in both arms were comparable in terms of age, height, and weight. A total of 277 participants completed the study, which included 135 (92.47%) in the biosimilar test arm and 142 (97.93%) in the reference arm. A total of 14

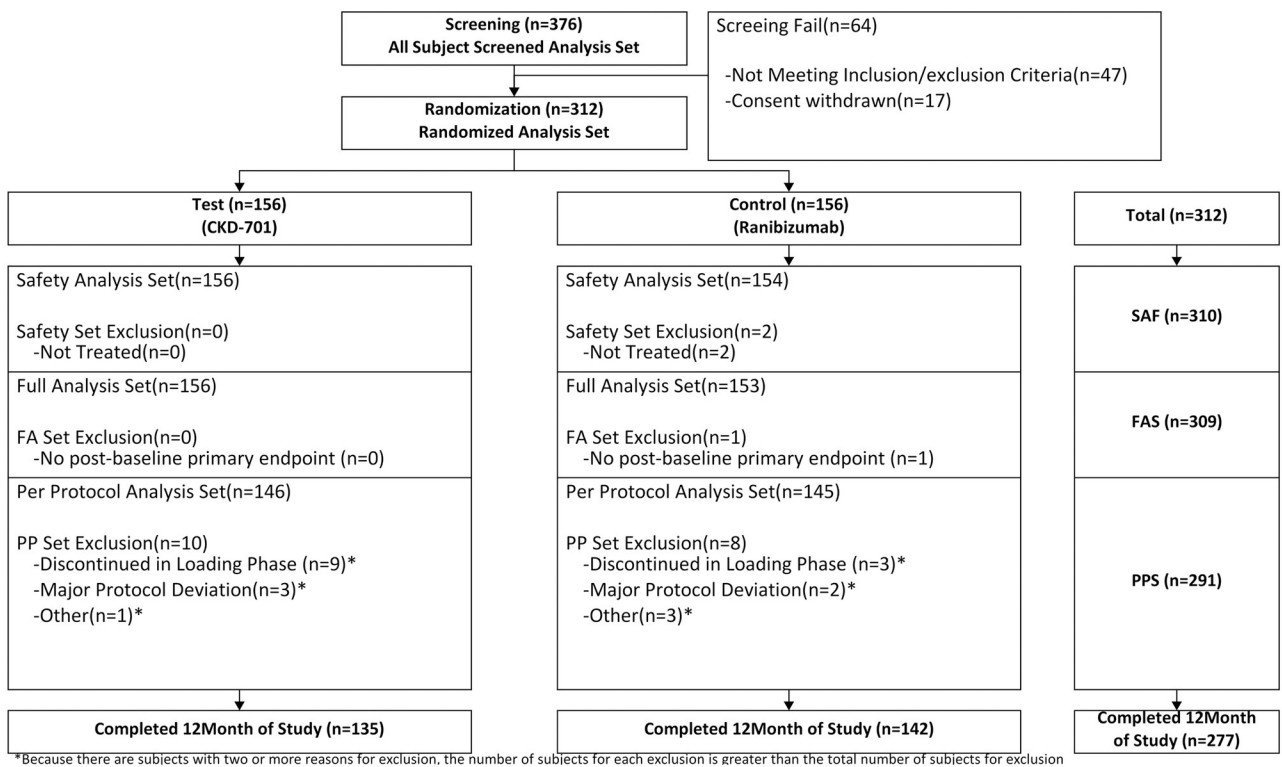

**Fig 1. Patient disposition.** SAF, safety analysis set; FAS, full analysis set; PPS, per-protocol set.

participants were discontinued from the study before loading phase completion, which included nine (6.16%) in the test arm and three (2.07%) in the reference arm. Details regarding patient disposition are presented in Table 1 and Fig 1.

## Efficacy assessment

**Severe vision loss at 3, 6, and 12 months compared to the baseline.** In the primary efficacy analysis, 143 (97.95%) patients lost <15 letters in the BCVA at 3 months in the CKD-701 arm, compared to 143 (98.62%) in the reference arm ($P = 0.67$). The proportion of patients with visual loss <15 letters in the BCVA at 6 and 12 months compared to the baseline was 97.92% (141/144 patients) and 95.52% (128/134 patients) in the CKD-701 arm, and 97.90% (140/143 patients) and 95.65% (132/138 patients) in the reference arm, respectively. No statistically significant difference was observed between the two arms ($P = 0.99$ and $0.98$ at 6 and 12 months, respectively) (Table 2).

**Change in best corrected visual acuity.** The results of the secondary efficacy analysis of 291 participants corresponding to the PP set are summarized in Table 3. The BCVA of the CKD-701 group was 59.38 letters (±13.10) at the baseline and 66.42 letters (±15.06), 66.08 letters (±16.55), and 66.07 letters (±18.09) at 3, 6, and 12 months after the first intravitreal injection, respectively (Fig 2). The BCVA of the reference group was 60.19 letters (±12.68) at the baseline and 66.37 letters (±13.66), 67.75 letters (±13.29), and 67.13 letters (±14.30) at 3, 6, and 12 months after administration, respectively. The PP set analysis of the proportion of patients from the CKD-701 group who exhibited visual improvement of ≥15 letters in the BCVA at 3, 6, and 12 months compared to the baseline, were 16.44% (24/146 patients), 21.53% (31/144 patients), and 24.63% (33/134 patients), respectively. In the reference group, the proportion

**Table 1. Baseline characteristics: Per protocol set.**

| | CKD-701 (n = 146) | Reference ranibizumab (n = 145) | Total (n = 291) |
|---|---|---|---|
| Age, years, mean (SD) | 72.18 (8.09) | 71.91 (8.79) | 72.05 (8.43) |
| Age group, years, n (%) | | | |
| 50–64 | 25(17.12) | 30(20.69) | 55(18.90) |
| >65 | 121(82.88) | 115(79.31) | 236(81.10) |
| Sex, female/male, n (%) | 57 (39.04) / 89 (60.96) | 64 (44.14) / 81 (55.86) | 121 (41.58) / 170 (58.42) |
| Height, cm, mean (SD) | 161.73 (9.37) | 160.47 (8.53) | 161.11 (8.97) |
| Weight, kg, mean (SD) | 64.93 (10.49) | 62.32 (11.35) | 63.63 (10.98) |
| Body mass index | 24.77 (3.03) | 24.12 (3.34) | 24.45 (3.20) |
| Duration of wet AMD, month, Mean (SD) | 0.37(1.20) | 1.51(7.05) | 0.94(5.07) |
| Binocular/Monocular, n (%) | 22(15.07) / 124(84.93) | 21(14.48) / 124(85.52) | 43(14.78) / 248(85.22) |
| Lesion size, disc area, % | 2.81(2.07) | 2.45(1.80) | 2.63(1.94) |
| Ratio of CNV area to lesion | 70.23(18.94) | 69.55(18.88) | 69.89(18.88) |
| Central retinal thickness, μm | 415.29(160.92) | 401.54(153.61) | 408.44(157.20) |

SD, standard deviation; CNV, choroidal neovascularization.

was 17.24% (25/145 patients), 19.58% (28/143 patients), and 21.01% (29/138 patients), respectively. Vision loss more than 15 letters was in 3 eyes (2.05%) vs 2 eyes (1.38%) at 3-month visit in CKD-701 arm and reference arm, respectively. No statistically significant difference was observed between the administration groups across all time points (all $P>0.05$).

**Table 2. Visual loss or gain from the baseline and at 3, 6, and 12 months: Per protocol analysis set.**

| | CKD-701 (n = 146) | Reference ranibizumab (n = 145) | Risk difference (SE) | 95% confidence interval | P-value |
|---|---|---|---|---|---|
| Participants who lost <15 letters in BCVA from the baseline, n (%) | | | | | |
| 3 months | 143(97.95) | 143(98.62) | -0.66(1.52) | -3.63, 2.32 | 0.67 |
| 6 months | 141(97.92) of 144* | 140(97.90) of [143]* | 0.02(1.69) | -3.29, 3.33 | 0.97 |
| 12 months | 128(95.52) of 134** | 132(95.65) of [138]** | -0.07(2.48) | -4.93, 4.79 | 0.98 |
| Participants who gain ≥15 letters in BCVA from the baseline, n (%) | | | | | |
| 3 months | 24(16.44) | 25(17.24) | -0.89(4.36) | -9.44, 7.67 | 0.84 |
| 6 months | 31(21.53)* | 28(19.58)* | 1.92(4.77) | -7.43, 11.26 | 0.69 |
| 12 months | 33(24.63)** | 29(21.01)** | 3.53(5.08) | -6.43, 13.48 | 0.49 |
| Participants who lost ≥ 15 letters in BCVA from baseline, n(%) | | | | | |
| 3 months | 3(2.05) | 2(1.38) | 0.66(1.52) | -2.32, 3.63 | 0.67 |
| 6 months | 3(2.08)* | 3(2.10)* | -0.02(1.69) | -3.33, 3.29 | 0.99 |
| 12 months | 6(4.48)** | 6(4.35)** | 0.07(2.48) | -4.79, 4.93 | 0.98 |

Cochran–Mantel–Haenszel Test (Covariate: PCV presence).

BCVA, best corrected visual acuity; SE, standard error.

*: Among the per protocol set participants, two were excluded from the analysis in the CKD-701 group (two dropouts during months 3–6), and two in the reference group (one dropout and one non-visit due to COVID-19 during months 3–6).

**: Among the per protocol set participants, 12 were excluded from the analysis in the CKD-701 group (two dropouts during months 3–6 and 10 dropouts during months 6–12), and seven in the reference group (one dropout during months 3–6 and six dropouts during months 6–12).

**Table 3. Changes in best corrected visual acuity (BCVA) at 3, 6, and 12 months: Per protocol analysis set.**

| | CKD-701 | | | | Reference ranibizumab | | | | Test results | | |
|---|---|---|---|---|---|---|---|---|---|---|---|
| | n | LS-Mean (SE) | Mean (SD) | *P*-value | n | LS-Mean (SE) | Mean (SD) | *P*-value | LS-Mean Diff (SE) | 95% CI | *P*-value |
| BCVA (letters) | | | | | | | | | | | |
| Baseline | 146 | | 59.38(13.10) | | 145 | | 60.19(12.68) | | | | |
| 3 months | 146 | | 66.42(15.06) | | 145 | | 66.37(13.66) | | | | |
| 6 months | 144* | | 66.08(16.55) | | 143* | | 67.75(13.29) | | | | |
| 12 months | 134** | | 66.07(18.09) | | 138** | | 67.13(14.30) | | | | |
| CFB at 3 months | 146 | 7.14(0.84) | 7.04(9.59) | <0.0001 [T] | 145 | 6.28(0.83) | 6.18(9.05) | <0.0001 [W] | 0.87(1.10) | -1.29, 3.02 | 0.43 |
| CFB at 6 months | 144 | 6.93(0.99) | 6.76(12.18) | <0.0001 [W] | 143 | 7.36(0.99) | 7.20(9.60) | <0.0001 [W] | -0.43(1.30) | -2.98, 2.13 | 0.743 |
| CFB at 12 months | 134 | 6.84(1.14) | 6.44(13.94) | <0.0001 [W] | 138 | 7.22(1.11) | 6.85(10.36) | <0.0001 [W] | -0.38(1.49) | -3.31, 2.54 | 0.80 |

CFB, change from baseline; BCVA, best corrected visual acuity; SD, standard deviation; SE, standard error; LS-Mean, least squares mean; CI, confidence interval.

[1] Paired t-test (T) or Wilcoxon signed rank test (W) result for change from baseline.

[2] 95% confidence interval for LS-mean differences between treatment groups.

[3] ANCOVA results for LS-mean comparison between administration groups (covariate: PCV presence or absence).

*: Among the per protocol set participants, two were excluded from the analysis in the CKD-701 group (two dropouts during months 3–6), and two in the reference group (one dropout and one non-visit due to COVID-19 during months 3–6).

**: Among the per protocol set participants, 12 were excluded from the analysis in the CKD-701 group (two dropouts during months 3–6 and 10 dropouts during months 6–12), and seven in the reference group (one dropout during months 3–6 and six dropouts during months 6–12).

**Central retinal thickness and the presence of intraretinal or subretinal fluid.** The 291 patients in the PP set from both treatment groups exhibited a sustained decrease in CRT after the first dose of the study medication. At the baseline, the CRT was 415.29 μm (±160.92) in the CKD-701 group and 401.54 μm (±153.61) in the reference group. According to the covariance analysis, the least square mean change in CRT at 1, 3, 6, and 12 months from the baseline among those in the CKD-701 arm were -97.36 μm (±10.07), -119.33 μm (±11.96), -118.16 μm (±12.39), and -123.50 μm (±13.35), respectively. Both administration groups showed significantly decreased CRT at all time points (all *P*<0.0001). Moreover, these values were not statistically different from the -98.87 μm (±10.07), -124.52 μm (±11.94), -128.81 μm (±12.36), and -130.84 μm (±13.08) in the reference arm (all *P*>0.05). The proportion of patients without intraretinal or subretinal fluid at 3, 6, and 12 months was also not different between the CKD-701 arm (54.11%, 48.61%, and 50.00%, respectively) and the reference arm (64.14%, 48.25%, and 52.17%, respectively) (all *P*>0.05).

**Number of injections during the study period and the PRN period.** During the entire clinical trial period, the average number of administrations of the investigational drug was 8.36 (±3.13) in the CKD-701 group and 8.26 (±2.92) in the reference group. No statistically significant difference was observed between the groups (*P* = 0.619). Furthermore, the mean number of administrations during the PRN phase was 5.76 (±2.76) in the CKD-701 group and 5.43 (±2.75) in the reference group. Similarly, there was no statistically significant difference between groups (*P* = 0.298).

## Immunogenicity and pharmacokinetics

There was a low cumulative incidence of antidrug antibodies up to 12 months, and this observation was similar between the treatment groups (CKD-701, 0 of 156 [0%]; reference, 2 of 156 [1.30%]). Similarly, when comparing the antibody incidence rate at each time point and the cumulative incidence rate up to 12 months during the entire clinical trial period, no statistically significant difference was observed between the administration groups at all time points (all *P*>0.05).

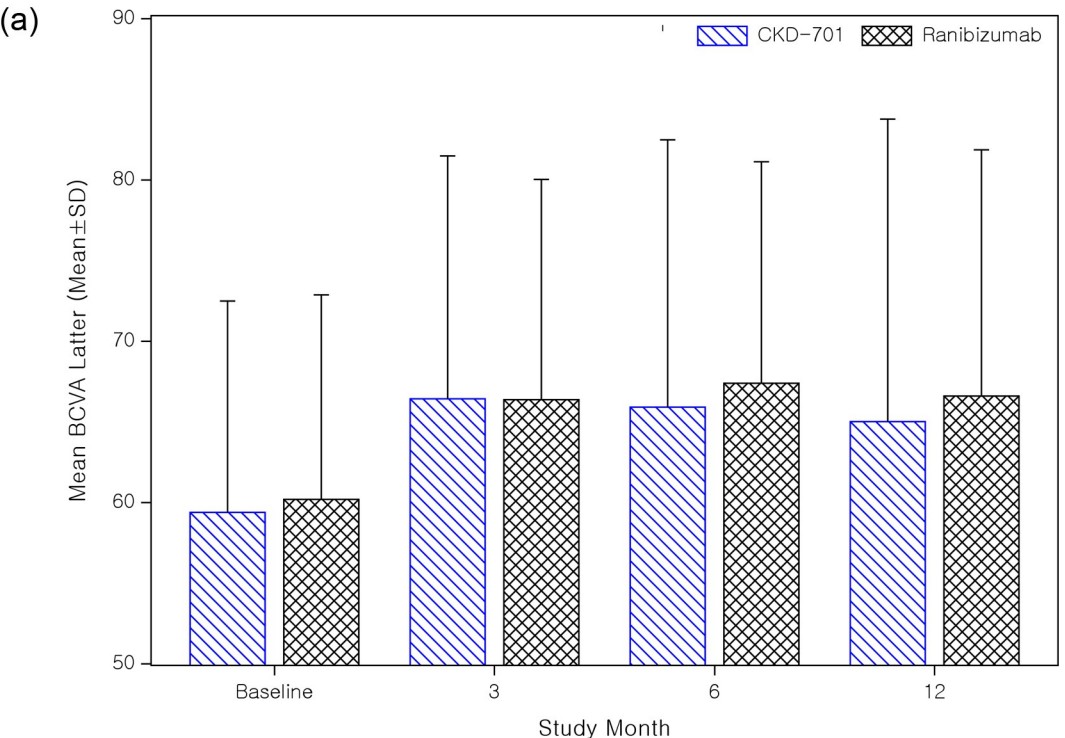

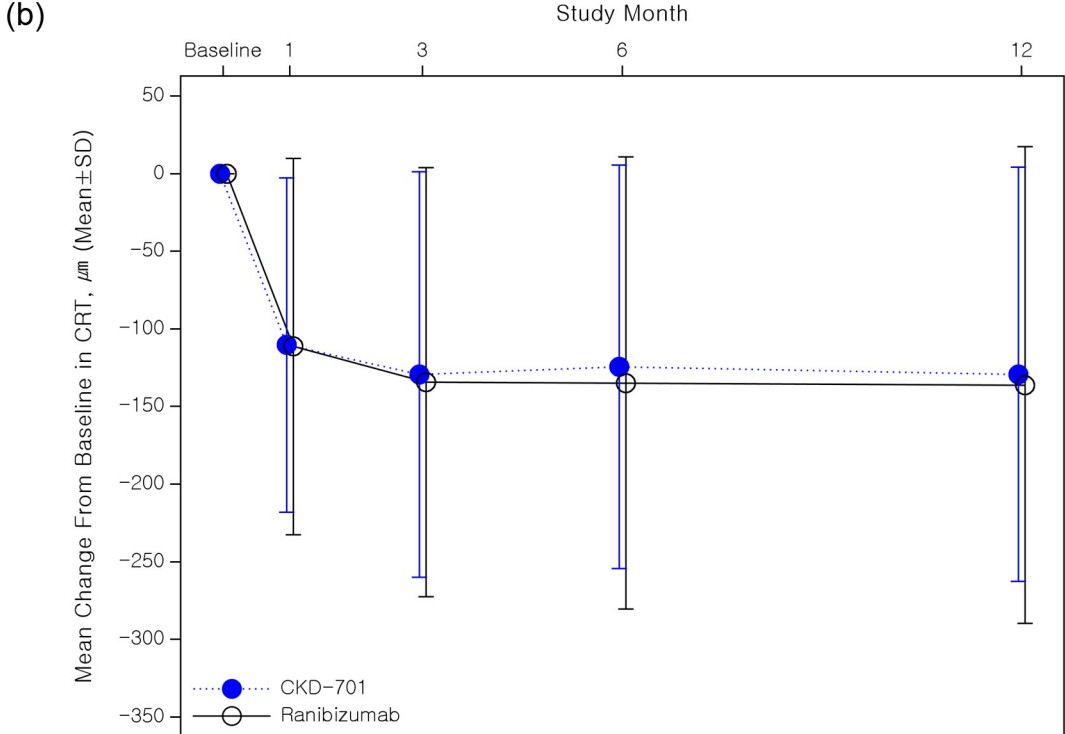

**Fig 2. Mean ± standard deviation (SD) change in best corrected visual acuity (BCVA) (upper) and central retinal thickness (lower) during the study (per-protocol analysis set).**

A total of 24 patients were enrolled for PK analysis; however, only the PK data from 22 patients were utilized because two patients were administered bevacizumab in their contralateral eye, which could affect the serum concentration of ranibizumab. Based on comparable $C_{max}$ and AUClast values, the PK profiles of the test and reference drugs were similar (Figs 3 and 4).

## Safety assessment

**Treatment-emergent adverse events.** The safety analysis set (Fig 1) consisted of all participants who received at least one administration of the study drug during the period after randomization. Adverse events that occurred during the study are listed in Table 4. Adverse events were recorded in 157 (50.65%) of 310 participants in the safety analysis set. Adverse reactions occurred in 80 patients (51.28%, 158 cases) in the CKD-701 group and 77 patients (50.00%, 153 cases) in the reference group, with the difference between the two groups not being statistically significant ($P = 0.821$). Six cases of serious adverse reactions related to the drug occurred among five patients. Nevertheless, there was no significant difference between the groups ($P = 0.683$). Serious adverse events (SAEs) that occurred after administration were reported in 16 patients (16 cases; 10.26%) in the CKD-701 arm and in 15 patients (17 cases; 9.74%) in the reference arm. However, there was no statistically significant difference between the groups ($P = 0.880$). The SAEs included coronavirus infection, pneumonia, bile duct stone, contusion, cerebral infarction, endophthalmitis, arthritis, and lumbar spinal stenosis, among others. Serious adverse drug reactions (SADR) that occurred after administration of the investigational drug included six cases among five patients (1.61%), including acute cardiac failure, myocardial infarction in the CKD-701 arm, and glaucoma, endophthalmitis, and carotid artery stenosis in the reference arm. However, there no statistically significant difference was observed between the CKD-701 group (two cases in two patients, 1.28%) and the reference group (four cases in three patients, 1.95%) ($P = 0.683$).

## Discussion

This study demonstrates the equivalence of the proposed CKD-701 product and the reference ranibizumab in terms of efficacy, safety, and immunogenicity in patients with nAMD when administered every month for the first 3 months (loading phase) followed by PRN treatment for 9 months. The difference in the proportion of patients who had 15 letters of vision loss between treatment arms, which was the primary endpoint, was 0.66% (95% confidence interval: -3.63–2.32) and met the pre-set equivalence range (+-11.5%). In addition, the secondary endpoints such as the changes in BCVA and CRT at 3, 6, and 12 months and the proportion of patients without intraretinal or subretinal fluid demonstrated that the treatment efficacy of CKD-701 was similar to that of the reference ranibizumab.

This prospective randomized clinical trial used a PRN dosing regimen rather than fixed dosing. The patients in this study received three loading injections, and additional injections were administered as needed. The PRN regimen used in this study was the standard of care when the study was initiated. The results showed that the PRN regimen was excellent, without visual or anatomic fluctuations. The PRN regimen in this study eventually reduced the number of injections by approximately 1.5. In addition, the PRN protocol has the advantage of being able to simultaneously compare the efficacy and durability of anti-VEGF products. This study may provide more strong evidence of similarity for CKD-701 compared to reference product than clinical trial based on fixed dosing.

In terms of safety, most of the adverse reactions reported during CKD-701 administration were similar to those reported in a previous clinical trial of ranibizumab, and the difference in

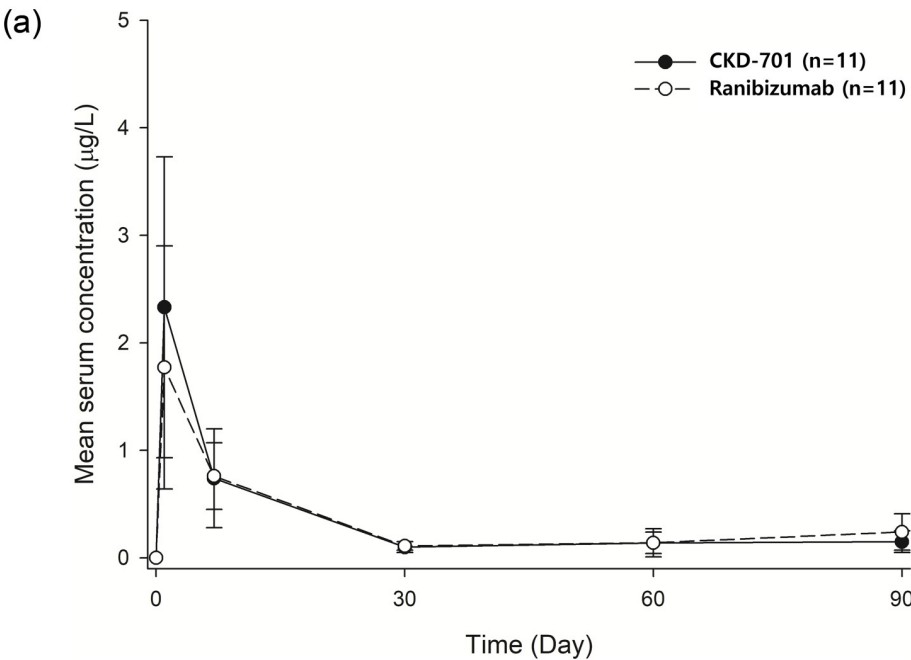

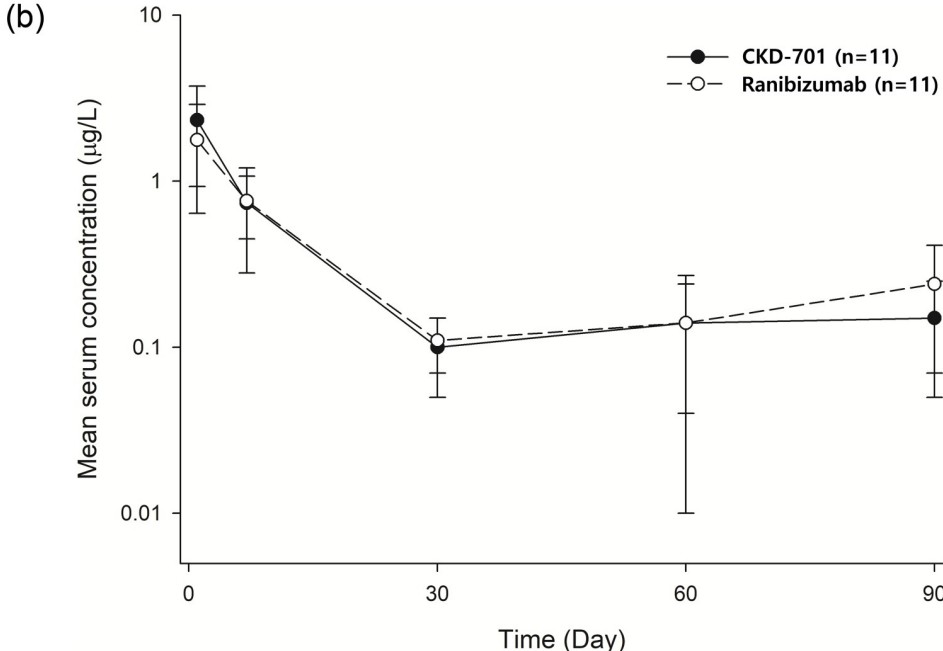

**Fig 3. Mean systemic ranibizumab concentration-time profile (left: Linear scale, right: Log scale).** Circles represent the mean and the error bar represents the standard deviation at each timepoint. Serum samples for the measurements of antibody concentration were obtained pre-dose (trough serum concentration [Ctrough]) and 24 to 72 hours post-dose (close to maximum serum concentration [$C_{max}$]) at day 0, 30, 60, and 90 and at day 1. The lower limit was 0.05 µg/L (Test: CKD-701, Reference: Lucentis).

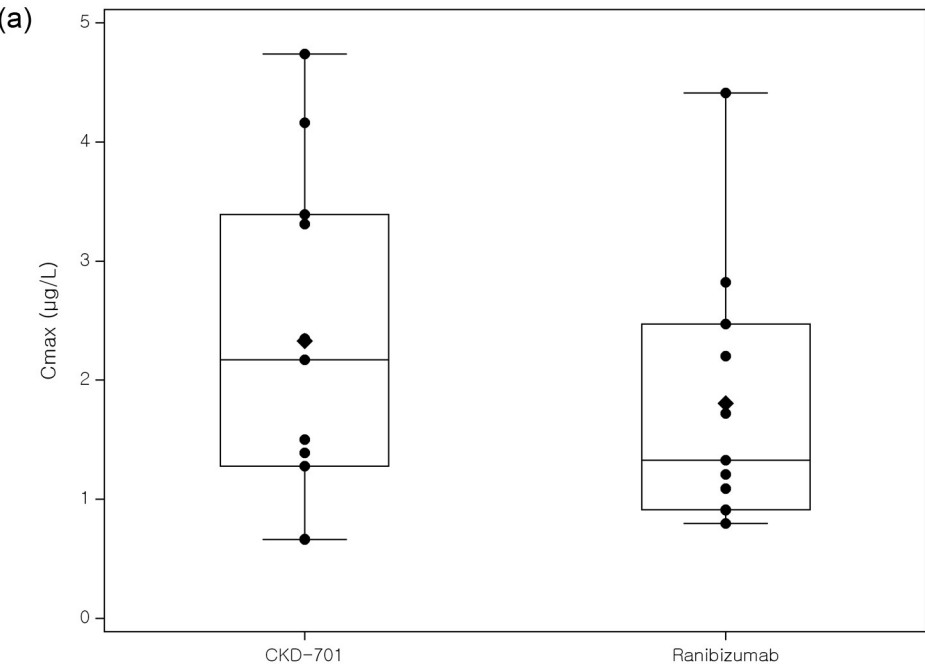

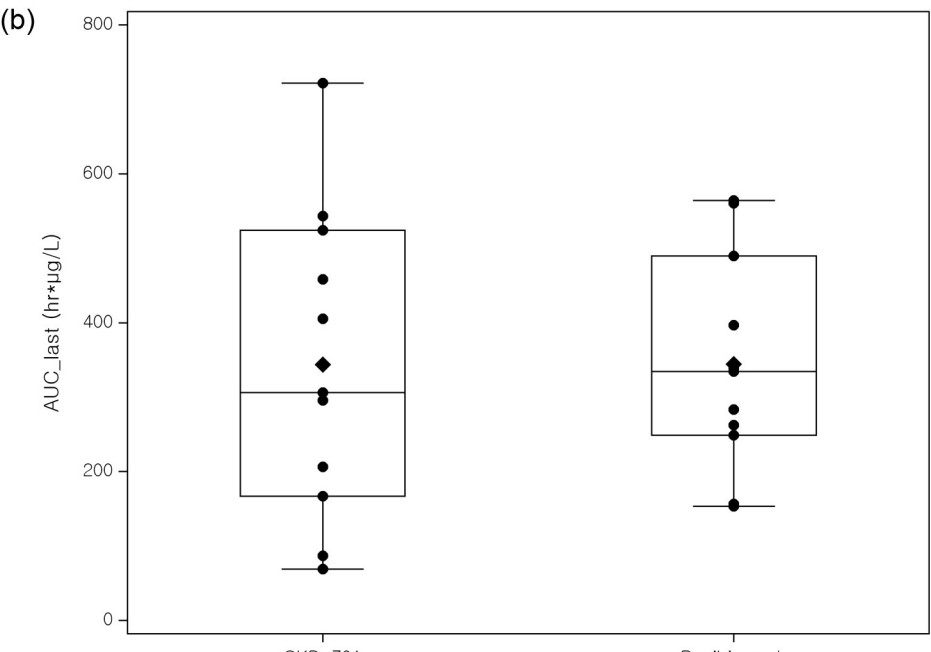

**Fig 4. Individual pharmacokinetic parameters of CKD-701 and ranibizumab (left: $C_{max}$, right: $AUC_{last}$).** $C_{max}$: Maximum plasma concentration, AUClast: The area under the concentration–time curve from time zero to the last quantifiable time point.

the incidence of adverse reactions between the treatment groups was not significant. Both $C_{max}$ and AUClast of the test and reference groups were similar, and other parameters, including apparent clearance, were comparable. Compared to the PK data from previous clinical studies, similar PK profiles of intravitreal ranibizumab injection were observed in the current

**Table 4. Adverse events recorded up to 12 months: Safety analysis set.**

| | CKD-701 (n = 156) | Reference ranibizumab (n = 154) | Total (n = 310) | *P*-value[1] |
|---|---|---|---|---|
| Participants with at least one TEAE, n (%) [case] | | | | |
| AE | 80(51.28)[158] | 77(50.00)[153] | 157(50.65)[311] | 0.82[c] |
| ADR | 28(17.95)[42] | 20(12.99)[30] | 48(15.48)[72] | 0.23[c] |
| SAE | 16(10.26)[16] | 15(9.74)[17] | 31(10.00)[33] | 0.88[c] |
| SADR | 2(1.28)[2] | 3(1.95)[4] | 5(1.61)[6] | 0.68[F] |
| AESI | 6(3.85)[6] | 8(5.19)[8] | 14(4.52)[14] | 0.57[c] |
| Ocular AE at test eye | 30(19.23)[47] | 27(17.53)[36] | 57(18.39)[83] | 0.70[c] |
| Ocular AE at fellow eye | 14(8.97)[15] | 10(6.49)[17] | 24(7.74)[32] | 0.41[c] |
| Non-Ocular AE | 52(33.33)[96] | 54(35.06)[100] | 106(34.19)[196] | 0.75[c] |
| Ocular AE in > 1.5% patients | | | | |
| Dry eye | 10(6.41)[11] | 5(3.25)[5] | 15(4.84)[16] | 0.19[C] |
| Neovascular AMD | 8(5.13)[8] | 4(2.60)[5] | 12(3.87)[13] | 0.25[C] |
| Vitreous floaters | 5(3.21)[5] | 1(0.65)[1] | 6(1.94)[6] | 0.21[F] |
| Conjunctival hemorrhage | 1(0.64)[1] | 4(2.60)[6] | 5(1.61)[7] | 0.21[F] |
| Retinal hemorrhage | 3(1.92)[3] | 2(1.30)[2] | 5(1.61)[5] | 1.00[F] |
| Cataract | 3(1.92)[3] | 1(0.65)[1] | 4(1.29)[4] | 0.62[F] |
| Eye pruritus | 3(1.92)[3] | - | 3(0.97)[3] | 0.25[F] |
| Non-ocular AE in >1.5% patients | | | | |
| Nasopharyngitis | 5(3.21)[6] | 3(1.95)[4] | 8(2.58)[10] | 0.72[F] |
| Pneumonia | 1(0.64)[1] | 3(1.95)[3] | 4(1.29)[4] | 0.379[F] |
| Back pain | 3(1.92)[3] | 1(0.65)[1] | 4(1.29)[4] | 0.62[F] |
| Gastritis | 1(0.64)[1] | 4(2.60)[4] | 5(1.61)[5] | 0.21[F] |
| Urticaria | 3(1.92)[4] | 2(1.30)[2] | 5(1.61)[6] | 1.00[F] |
| Hypertension | 2(1.28)[2] | 3(1.95)[3] | 5(1.61)[5] | 0.68[F] |

[1] Test results for comparison between treatment groups (Chi-square test [C] or Fisher's exact test [F]).

TEAE, treatment-related adverse events; AE, adverse event; ADR, adverse drug reaction; SAE, serious adverse event; SADR, serious adverse drug reaction; AESI, adverse event of special interest; AMD, age-related macular degeneration.

study. Considering the range of serum ranibizumab after intravitreal ranibizumab injection, systemic exposure to intravitreal ranibizumab was negligible. Immunogenicity analysis also showed that the incidence of neutralizing antibodies was not significantly different between the administration groups across all time points.

Currently, research on biosimilars for retinal diseases are being actively conducted [11]. Among them, ranibizumab biosimilars have already received national approval and are being used in patients [16,17]. For biosimilars other than ranibizumab, the results of a phase 3 study have been recently published and are expected to be used in the near future [12,13]. There are few reports about ophthalmic biosimilars experience in the real world; hence, there is limited information on whether biosimilars may cause unexpected and rare side effects, especially after intravitreal injections. For example, sterile intraocular inflammation, which is reportedly associated with the manufacturing process of the ranibizumab biosimilar, has also been documented after market approval. However, evidence acquired over several years of clinical experience has indicated that biosimilars are as safe and effective in their approved indications as other biological drugs. In a recent study, Sharma et al. reviewed charts of treatment-naïve patients with nAMD who have received a ranibizumab biosimilar (Razumab™, Intas

Pharmaceuticals Ltd, India), which was approved in 2015 [17]. They reported that none of the patients had ocular or systemic adverse events, including clinical signs of immunogenicity. In other studies, no additional safety issues were reported [18,19] To date, there are no published results supporting the hypothesis that intravitreal injection of a biosimilar may cause unexpected immunological adverse reactions.

In general, patients with nAMD require repetitive anti-VEGF injections to achieve better outcomes. However, the relatively high price of original biologics may limit the affordability of this treatment. By contrast, ranibizumab biosimilars, including the current study drug, can expand accessibility to anti-VEGF treatment and improve eventual treatment outcomes in the context of public ophthalmic healthcare. The development of biosimilars is expected to lower the cost of treatment for retinal disease through competition with the original drugs [20]. It has also been suggested that whether the introduction of a biosimilar could lower the cost of treating nAMD depends on off-label bevacizumab [13,21]. Sheth et al. reported that a third of Indian vitreoretinal specialists believe that the price of the ranibizumab biosimilar is a reasonable replacement for off-label bevacizumab in India [22]. It was also shown that a ranibizumab biosimilar (Razumab™) rapidly grew its user base [21]. Although the present study was a multi-center study, it was conducted in only one country (South Korea), which may limit the applicability of the study results to patients with different ethnic backgrounds. In addition, since the type of nAMD and the genetic background of the participants were similar between groups, there may have been less variation in treatment outcomes. However, treatment outcomes with reference ranibizumab were not significantly different between patients of different ethnic backgrounds, suggesting that there would be little difference in treatment outcomes across countries [23].

In conclusion, the biosimilar CKD-701 had comparable efficacy and safety profiles to that of the reference ranibizumab, as evident from the efficacy variables analyzed between the treatment arms. Pro re nata dosing regimen can stabilize wet AMD activity as well as reducing treatment burden. Therefore, based on this comparability, it is proposed that the indigenous ranibizumab biosimilar may be considered as a viable alternative to the reference innovator product in patients with nAMD.

## Supporting information

**S1 Checklist.**
(DOC)

**S1 File. Protocol file of this clinical trial.**
(PDF)

**S2 File. Minimal study data.** This file contains data to reach the conclusion of this study.
(ZIP)

## Acknowledgments

The authors would like to thank the investigators who contributed to the generation of the study data: Dr. Dae Young Lee, Dr. Young-Hoon Park, Dr. Sung Pyo Park, Dr. Jae Pil Shin, Dr. Seung-Young Yu, Dr. Yu Cheol Kim, Dr. Cheol Min Yun, Dr. Yong Sung You, Dr. Yun-Young Kim, Dr. Se Woong Kang, Dr. Joo Yong Lee, Dr. JuYeong Shin, Dr. Sung Jin Lee, Dr. Kihwang Lee, Dr. Min Sagong, Dr. Hyun Woong Kim, Dr. Hee Seung Chin, Dr. Jin Gu Jeong, Dr. Jung Yeul Kim, Dr. Joonhong Sohn, Dr. Kayoung Yi, Dr. Kang Yeun Pak, and Dr. IkSoo Byon.

## Author Contributions

**Conceptualization:** Hyeong Gon Yu.

**Data curation:** Chang Ki Yoon.

**Formal analysis:** Chang Ki Yoon, Jaeryung Oh.

**Methodology:** Jaeryung Oh, Kyung-Sang Yu, Hyeong Gon Yu.

**Project administration:** Jaeryung Oh, Un Chul Park.

**Supervision:** Jaeryung Oh, Un Chul Park, Hyeong Gon Yu.

**Validation:** Hyeong Gon Yu.

**Writing – original draft:** Chang Ki Yoon, Jaeryung Oh, Kunho Bae.

**Writing – review & editing:** Kyung-Sang Yu.

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
