## [Decision Letter · Decision Letter 0]

21 Jun 2022

PONE-D-21-38254Efficacy and safety of a new ranibizumab biosimilar CKD-701 using a pro re nata treatment regimen in neovascular age-related macular degeneration: a phase 3 randomized clinical trialPLOS ONE

Dear Dr. Yu,

Thank you for submitting your manuscript to PLOS ONE. After careful consideration, we feel that it has merit but does not fully meet PLOS ONE’s publication criteria as it currently stands. Therefore, we invite you to submit a revised version of the manuscript that addresses the points raised during the review process.

We look forward to receiving your revised manuscript.

Kind regards,

Demetrios G. Vavvas

Academic Editor

PLOS ONE

**Journal requirements:**

2. Thank you for submitting your clinical trial to PLOS ONE and for providing the name of the registry and the registration number. The information in the registry entry suggests that your trial was registered after patient recruitment began. PLOS ONE strongly encourages authors to register all trials before recruiting the first participant in a study.

a) your reasons for your delay in registering this study (after enrolment of participants started);

b) confirmation that all related trials are registered by stating: “The authors confirm that all ongoing and related trials for this drug/intervention are registered”.

3. We note that the original protocol file you uploaded contains a confidentiality notice indicating that the protocol may not be shared publicly or be published. Please note, however, that the PLOS Editorial Policy requires that the original protocol be published alongside your manuscript in the event of acceptance. Please note that should your paper be accepted, all content including the protocol will be published under the Creative Commons Attribution (CC BY) 4.0 license, which means that it will be freely available online, and any third party is permitted to access, download, copy, distribute, and use these materials in any way, even commercially, with proper attribution.

Therefore, we ask that you please seek permission from the study sponsor or body imposing the restriction on sharing this document to publish this protocol under CC BY 4.0 if your work is accepted. We kindly ask that you upload a formal statement signed by an institutional representative clarifying whether you will be able to comply with this policy. Additionally, please upload a clean copy of the protocol with the confidentiality notice (and any copyrighted institutional logos or signatures) removed.

4. We note that the original protocol that you have uploaded as a Supporting Information file contains an institutional logo. As this logo is likely copyrighted, we ask that you please remove it from this file and upload an updated version upon resubmission.

“The study was funded by the sponsor, CKD. Ltd (Chong Kun Dang). Study design, collection and analysis of data were performed by the respective groups of the sponsor.

Korean Association of Retinal Degeneration also funded this study and participated as image reading center (Seoul Reading Center).”

**Additional Editor Comments:**

First, my apologies for taking some more time than usual due to my personal illness.

Both reviewers found this study rather well done and have some suggestions to make it more clear.

Reviewers' comments:

Reviewer's Responses to Questions

**Comments to the Author**

1. Is the manuscript technically sound, and do the data support the conclusions?

Reviewer #1: Yes

Reviewer #2: Yes

2. Has the statistical analysis been performed appropriately and rigorously? 

Reviewer #1: Yes

Reviewer #2: Yes

3. Have the authors made all data underlying the findings in their manuscript fully available?

Reviewer #1: No

Reviewer #2: Yes

4. Is the manuscript presented in an intelligible fashion and written in standard English?

Reviewer #1: Yes

Reviewer #2: Yes

5. Review Comments to the Author

Reviewer #1: Thank you for submitting this interesting work to PLOS ONE.

This is overall a well-designed study on investigating the efficacy, safety and immunogenicity of the CKD-701 biosimilar and a nicely written manuscript.

Here are my comments and suggestions.

In the Methods section you mention the “key exclusion criteria”. Are they any additional ones that were either applied to the study eye or the fellow eye such as presence of other concurrent non-retinal pathologies? Also, any systemic diseases or medications? If so, I would highly suggest you include those as well (supplemental material).

Regarding the PRN injection criteria, I would recommend you comment on whether all of these or any combination or even only one criterium could be met in order to proceed with the injection.

Was angiography (FA or ICGA) also performed at other points or only at baseline?

What about the patients who lost >15 letters?

Did you perform a subanalysis on the fellow eyes?

Thank you.

Reviewer #2: Abstract: line 23. The 3 months compared to baseline - when is baseline relative to the initial 3-month loading intraocular injections, and when is the 3 months? Is it before the PRN dosing? This is a little unclear at the moment.

lines 28, 30, 32, 34, 181, 186, etc: p-values need only be to 2 decimal places at this value.

Introduction: line 46: what is "wet" AMD?

line 66 - no need for a "," after "Although, ".

Material and methods

The primary endpoint is whether the patient lost fewer than 15 letters of visual acuity at 3 months compared to the baseline value.

I presume this is the quantity used in the sample size justification in line 137-139, but that doesn't seem to be the situation. What is actually the rate of 30% difference between the ranibizumab and placebo group? It is not currently clear why this sample size was determined - perhaps the rates in the groups as well as the difference?

(is there some confusion with participants who lost<15 letters and participants who gain>= 15 letters?)

line 192: after first administration?

table 2,3,4: p-values needn't be to 4 decimal places. 2 would be satisfactory.

Figures refer to the drug as Lucentis - this is not in the text anywhere - ensure there is no ambiguity as to whether these figures are using the same treatment as the text in the study.

Was any consideration given to fitting mixed models to the data to account for the differing time points among the same patients in the models? This could provide an average treatment effect for the difference between treatment arms over time - rather than just the 3 month after study entry timepoint.

In general the time points could refer to time since randomisation to reduce uncertainty regarding when the timepjoint was.

6. PLOS authors have the option to publish the peer review history of their article (what does this mean?). If published, this will include your full peer review and any attached files.

Reviewer #1: No

Reviewer #2: No

---

## [Author Response · Author response to Decision Letter 0]

13 Aug 2022

Reviewer #1: Thank you for submitting this interesting work to PLOS ONE.

This is overall a well-designed study on investigating the efficacy, safety and immunogenicity of the CKD-701 biosimilar and a nicely written manuscript.

Here are my comments and suggestions.

1. In the Methods section you mention the “key exclusion criteria”. Are they any additional ones that were either applied to the study eye or the fellow eye such as presence of other concurrent non-retinal pathologies? Also, any systemic diseases or medications? If so, I would highly suggest you include those as well (supplemental material).

Response: We appreciate your review and comments. We excluded patients having ocular disease other than AMD, systemic infection and taking systemic medication which can result in retinal adverse events. Detailed inclusion and exclusion criteria are described in supplementary protocol. (Page 5 line 99 and Supporting file (S1 file) page 4)

2. Regarding the PRN injection criteria, I would recommend you comment on whether all of these or any combination or even only one criterium could be met in order to proceed with the injection.

Response: Clinical trial drug was re-administered when any one of the PRN administration criteria were met. We corrected the PRN treatment protocol in the manuscript clearly. (Page 6 line 124) 

3. Was angiography (FA or ICGA) also performed at other points or only at baseline?

Response: Fluorescein angiography was also performed at the end of the study.

4. What about the patients who lost >15 letters?

Response: We added this data, who lost more than 15 letters, in the Table 2. 

5. Did you perform a subanalysis on the fellow eyes?

Response: Thank you for your comment. We are preparing the post hoc analysis about this study data including fellow eye characteristics in next publication.

Reviewer #2: 

1. Abstract: line 23. The 3 months compared to baseline - when is baseline relative to the initial 3-month loading intraocular injections, and when is the 3 months? Is it before the PRN dosing? This is a little unclear at the moment.

Response: I appreciate your comment. We agree with your opinion that the description is unclear. The secondary outcome time point was 3 months after the baseline visit, one month after last loading injection and before PRN dosing. We corrected the description more clearly. (Page 2 28)

2. lines 28, 30, 32, 34, 181, 186, etc: p-values need only be to 2 decimal places at this value.

Response: We corrected the p-values to be 2 decimal. Thank you for your comment

3. Introduction: line 46: what is "wet" AMD?

Response: Thank you for your comment. We replaced ‘wet AMD’ to neovascular AMD. 

4. line 66 - no need for a "," after "Although, ".

Response: We deleted the comma. Thank you for kind comment.

5. 

Material and methods

The primary endpoint is whether the patient lost fewer than 15 letters of visual acuity at 3 months compared to the baseline value.

I presume this is the quantity used in the sample size justification in line 137-139, but that doesn't seem to be the situation. What is actually the rate of 30% difference between the ranibizumab and placebo group? It is not currently clear why this sample size was determined - perhaps the rates in the groups as well as the difference?

(is there some confusion with participants who lost<15 letters and participants who gain>= 15 letters?)

Response: I appreciate your comment. In pivotal study of Ranibizumab for wetAMD, MARINA study, proportion of patients who lost fewer than 15 letters were 94.6% in 0.5 mg ranibizumab and 62.2% in sham injection at 12 months. Difference of proportion was more than 30% in this pivotal study. Our primary outcome was measured at 3 months because drug response is most sensitive at this time and the improvement of vision maintained through 1 year. We corrected the reference as MARINA study from ANCHOR study.(Reference 15, Page 7 line 149)

6. line 192: after first administration?

Response: Thank you for your comment. The expression, ‘after first administration’, means after the first intravitreal injection because the first injection was performed on the day of randomization. We corrected this description not to make confusion. (Page 10 line 203)

7. table 2,3,4: p-values needn't be to 4 decimal places. 2 would be satisfactory.

Response: We corrected the p-values to 2-decimal value.

8. Figures refer to the drug as Lucentis - this is not in the text anywhere - ensure there is no ambiguity as to whether these figures are using the same treatment as the text in the study.

Thank you for your comment. I understand your confusion. Pharmacokinetics sample was performed in 12 patients of each CKD-701 (study drug) and Ranibizumab (reference drug). We replaced Lucentis to Ranibizumab to clarify.

9. Was any consideration given to fitting mixed models to the data to account for the differing time points among the same patients in the models? This could provide an average treatment effect for the difference between treatment arms over time - rather than just the 3 month after study entry timepoint.

In general the time points could refer to time since randomisation to reduce uncertainty regarding when the time point was.

Response: Thank you for your comment. We agree with you. We added the description regarding the baseline time point. (Page 6 line 129)

---

## [Decision Letter · Decision Letter 1]

4 Sep 2022

PONE-D-21-38254R1Efficacy and safety of a new ranibizumab biosimilar CKD-701 using a pro re nata treatment regimen in neovascular age-related macular degeneration: a phase 3 randomized clinical trialPLOS ONE

Dear Dr. Yu,

Thank you for submitting your manuscript to PLOS ONE. After careful consideration, we feel that it has merit but does not fully meet PLOS ONE’s publication criteria as it currently stands. Therefore, we invite you to submit a revised version of the manuscript that addresses the points raised during the review process.

 There is one more minor change that needs to be addressed before final acceptance

We look forward to receiving your revised manuscript.

Kind regards,

Demetrios G. Vavvas

Academic Editor

PLOS ONE

Journal Requirements:

Reviewers' comments:

Reviewer's Responses to Questions

**Comments to the Author**

1. If the authors have adequately addressed your comments raised in a previous round of review and you feel that this manuscript is now acceptable for publication, you may indicate that here to bypass the “Comments to the Author” section, enter your conflict of interest statement in the “Confidential to Editor” section, and submit your "Accept" recommendation.

Reviewer #1: All comments have been addressed

Reviewer #2: (No Response)

2. Is the manuscript technically sound, and do the data support the conclusions?

Reviewer #1: Yes

Reviewer #2: Yes

3. Has the statistical analysis been performed appropriately and rigorously? 

Reviewer #1: Yes

Reviewer #2: Yes

4. Have the authors made all data underlying the findings in their manuscript fully available?

Reviewer #1: (No Response)

Reviewer #2: Yes

5. Is the manuscript presented in an intelligible fashion and written in standard English?

Reviewer #1: (No Response)

Reviewer #2: Yes

6. Review Comments to the Author

Reviewer #1: Thank you for submitting your revised manuscript and for adequately addressing all main questions and comments. This is an interesting study and has been a great improvement in the quality of the manuscript. Thank you.

Reviewer #2: In the track changed version of the manuscript line 154.

I think some confusion has crept in with the sample size justification. Previously the statistic was 30% (95%CI: 23-37%). This was indicating that the estimate of the difference between two groups was 30%, with 95% confidence that the true estimate is between 23% and 37%. This is OK.

I think now the authors were intending to state that the separate groups had rates of 94.6% and 62.2% - and the difference was 32%. This is not currently what the sentence says - the 95% CI is incorrect. Please address prior to acceptance of manuscript.

7. PLOS authors have the option to publish the peer review history of their article (what does this mean?). If published, this will include your full peer review and any attached files.

Reviewer #1: No

Reviewer #2: No

---

## [Author Response · Author response to Decision Letter 1]

17 Sep 2022

Reviewer #2: In the track changed version of the manuscript line 154.

I think some confusion has crept in with the sample size justification. Previously the statistic was 30% (95%CI: 23-37%). This was indicating that the estimate of the difference between two groups was 30%, with 95% confidence that the true estimate is between 23% and 37%. This is OK.

I think now the authors were intending to state that the separate groups had rates of 94.6% and 62.2% - and the difference was 32%. This is not currently what the sentence says - the 95% CI is incorrect. Please address prior to acceptance of manuscript.

Thank you for pointy comment. I totally agree with you on this issue. Statistician of the study team referenced the value in the official document from U.S. FDA (Statistical Review and Evaluation Clinical Studies BLA/Serial Number 125156, url: https://www.accessdata.fda.gov/drugsatfda_docs/nda/2006/125156s0000_Lucentis_StatR.pdf) instead of publication in NEJM. This document is about pivotal study “Ranibizumab for Neovascular Age-Related Macular Degeneration”, also called MARINA study. This document can be downloaded freely. Table 5 of this document described that Loss of <15 letters difference (95% CI) is 30% (23, 37%). This confusion might not have been made if I had referenced this document on the first submission. I replaced the reference correctly at this time. (Page 7 Line 147)

---

## [Editor Report · Decision Letter 2]

20 Sep 2022

Efficacy and safety of a new ranibizumab biosimilar CKD-701 using a pro re nata treatment regimen in neovascular age-related macular degeneration: a phase 3 randomized clinical trial

PONE-D-21-38254R2

Dear Dr. Yu,

We’re pleased to inform you that your manuscript has been judged scientifically suitable for publication and will be formally accepted for publication once it meets all outstanding technical requirements.

Kind regards,

Demetrios G. Vavvas

Academic Editor

PLOS ONE
---

## [Editor Report · Acceptance letter]

3 Nov 2022

PONE-D-21-38254R2 

Efficacy and safety of a new ranibizumab biosimilar CKD-701 using a pro re nata treatment regimen in neovascular age-related macular degeneration: a phase 3 randomized clinical trial 

Dear Dr. Yu:

I'm pleased to inform you that your manuscript has been deemed suitable for publication in PLOS ONE. Congratulations! Your manuscript is now with our production department. 

Kind regards, 

on behalf of

Prof. Demetrios G. Vavvas 

Academic Editor

PLOS ONE